# Indigeneity and Likelihood of Discharge to Psychiatric Hospital in an Australian Deliberate Self-Poisoning Hospital-Treated Cohort

**DOI:** 10.3390/ijerph191912238

**Published:** 2022-09-27

**Authors:** Katie McGill, Amir Salem, Tanya L. Hanstock, Todd R. Heard, Leonie Garvey, Bernard Leckning, Ian Whyte, Andrew Page, Greg Carter

**Affiliations:** 1School of Medicine and Public Health, University of Newcastle, Newcastle, NSW 2308, Australia; 2Healthy Minds, Hunter Medical Research Institute, Newcastle, NSW 2308, Australia; 3Mental Health-Research, Evaluation and Dissemination (MH-READ), Hunter New England Local Health District, Newcastle, NSW 2298, Australia; 4School of Psychological Sciences, University of Newcastle, Newcastle, NSW 2308, Australia; 5Wiyillin Ta Child and Adolescent Mental Health Service, Hunter New England Local Health District, Newcastle, NSW 2300, Australia; 6Aboriginal Health, Hunter New England Local Health District, Wallsend, NSW 2287, Australia; 7Black Dog Institute, University of New South Wales, Sydney, NSW 1466, Australia; 8Menzies School of Health Research, Charles Darwin University, Darwin, NT 0811, Australia; 9Calvary Mater Newcastle, Newcastle, NSW 2298, Australia; 10Translational Health Research Institute, Western Sydney University, Penrith, NSW 2751, Australia

**Keywords:** deliberate self-poisoning, deliberate self-harm, psychiatric after-care, indigenous

## Abstract

Hospital-treated self-harm rates for Aboriginal and Torres Strait Islander (Indigenous) people are at least double those for other Australians. Despite this, limited research has explored the relationship between Indigeneity and the clinical management of hospital-treated deliberate self-harm. A retrospective clinical cohort study (2003–2012) at a regional referral centre (NSW) for deliberate self-poisoning was used to explore the magnitude and direction of the relationship between Indigeneity and discharge destination (psychiatric hospital vs. other) using a series of logistic regressions. There were 149 (4%) Indigenous and 3697 (96%) non-Indigenous deliberate self-poisoning admissions during the study period. One-third (31%) were referred to the psychiatric hospital at discharge; Indigenous 21% (*n* = 32) vs. non-Indigenous 32% (*n* = 1175). Those who identified as Indigenous were less likely to be discharged to the psychiatric hospital, OR 0.59 (0.40–0.87) at the univariate level, with little change after sequential adjustment; and AOR 0.34 (0.21–0.73) in the fully adjusted model. The Indigenous cohort had a lower likelihood of psychiatric hospital discharge even after adjustment for variables associated with discharge to the psychiatric hospital highlighting the need for further investigation of the reasons accounting for this differential pattern of clinical management and the effectiveness of differential after-care allocation.

## 1. Introduction

Aboriginal and Torres Strait Islander people in Australia (referred to in this study as Indigenous people) have the oldest living culture, flourishing with diversity and flexibility for over 50,000 years [1]. The impact of colonisation on Australian Indigenous communities is significant. Indigenous Australians live with the effects of historical and inter-generational trauma, disconnection because of loss of cultural identity, disruption of spiritual connection to land, being taken away from family, the ripple effects of the Stolen Generation and the impact of ongoing racism and discrimination [2,3]. Indigenous Australians also experience poorer mental and physical health compared to other Australians, including consistently higher rates of suicide, self-harm and suicidal ideation [3,4,5,6].

Hospital-treated self-harm refers to self-poisoning or self-injury that has resulted in hospital treatment [7]. It is the strongest independent risk factor associated with later suicide [8]. Whilst there is wide regional variability, in Australia, the reported rates of hospital-treated self-harm for Indigenous people are at least double (384 per 100,000) those for non-Indigenous Australians nationally (131 per 100,000) [5,9]; and these rates are acknowledged underestimates [5]. There has generally been an upward trend in reported hospital-treated self-harm rates since 2005, particularly for young Indigenous females [10]. For both Indigenous and non-Indigenous people, the most common type of hospital-treated self-harm is deliberate self-poisoning, accounting for over 80% of Australian self-harm hospital admissions each year, with other types of self-injury (such as cutting, burning) and methods associated with more medically serious or lethal outcomes (such as hanging, jumping and shooting) making up a smaller proportion of self-harm admissions [11]. While rates of hospital-treated self-harm are highest among 15–24-year-olds for both Indigenous and non-Indigenous Australians; for non-Indigenous Australians, rates decline for older age groups, whereas the same age-related reduction in older age groups is not evident for Indigenous Australians [11].

To date, only a few peer-reviewed studies have examined hospital presentations for self-harm by Indigenous people within Australia. Studies conducted in the Northern Territory (whole jurisdiction) [12,13] and Western Australia (Kimberley region) [14,15] have identified much higher self-harm and suicidal presentation rates for the Indigenous patient cohort [12,13], as well as a greater proportion presenting with non-self-poisoning methods of self-harm and other differences in presenting clinical characteristics (e.g., psychiatric diagnoses, psychosocial circumstances) [12,13], previous contact with mental health services [14] and trends in rates over time [12]. While these findings are consistent with official institutional reporting (e.g., [11]) and reviews [16], it is not clear how generalisable the findings are because the study locations are unique settings (regional or remote locations with Indigenous people representing a greater proportion of the population compared to other settings) and the patterns of findings may not be directly relevant to larger metropolitan centres, places with less remote catchment areas or Indigenous communities with different characteristics.

For all populations, most episodes of hospital-treated self-harm result in a discharge directly home; however, a substantial minority require more acute care, resulting in admission to psychiatric units [17]. The efficacy of psychiatric hospitalisation in reducing self-harm repetition or suicides is largely unexamined [18]. One systematic review identified that suicide prevention interventions delivered in psychiatric hospital settings were associated with larger effect sizes in reducing suicide attempt repetition and deaths compared to interventions delivered in other settings (e.g., outpatient mental health settings, emergency room settings, community-level settings, etc., [19]). However, admission to psychiatric hospital is a significant risk factor for later suicide death [20] and some have argued that the hospitalisation experience itself may contribute to this increased risk (e.g., [21]), particularly with regard to the range of acknowledged risks of harm associated with psychiatric admission [18].

Overall, rates of hospital admissions for specialised psychiatric care (for any cause) for Indigenous people in Australia are approximately double those of non-Indigenous Australians (154 vs. 63 per 10,000) [22]. While there are some signs that the hospitalisation disparities associated with Indigeneity can be accounted for by differing socio-demographic and clinical characteristics [12,23], there are also indications that Indigenous patients may experience more restrictive care within psychiatric units (e.g., higher rates of seclusion and restraint) [24] and experience poorer outcomes (e.g., higher 1-month readmission rate for acute mental health short stay care) [25].

There is even less information about rates or experiences of psychiatric hospitalisation after hospital-treated self-harm for Indigenous people in Australia. In one example from the Northern Territory, 43% of all ED presentations for suicidal thoughts and related behaviours by Indigenous people were admitted for further care; this included admissions for both further medical and psychiatric care [12]. More broadly, these patterns of hospital service use occur in the context of limited community trust and wariness of government institutions, including health services (due to experiences and impacts of institutional programs and intervention, e.g., that led to the Stolen Generation [26,27]), stigma and silence about mental health problems and suicide (e.g., [28,29,30,31]) and experiences of stereotyping, assumptions and racism, including within healthcare settings [32].

Thus, while Indigenous people have much higher rates of hospitalisation for self-harm and psychiatric admissions, we know little about the relative benefits and harms of psychiatric hospitalisation generally, particularly in the context of hospital-treated deliberate self-harm. If potential iatrogenic effects associated with psychiatric hospitalisation differentially and adversely impact Indigenous patients, it is possible that psychiatric hospitalisation itself may play a role in widening (rather than closing) the gap in suicide outcomes for Indigenous and non-Indigenous Australians.

Within this context, a better understanding of the characteristics and clinical management of hospital-treated self-harm (specifically self-poisoning) by Indigenous people is an important first step toward ensuring general hospital, psychiatric hospital and after-care for Indigenous people is clinically effective, culturally sensitive, appropriate and tailored to context. Nationally, Indigenous hospital-treated self-harm rates have been increasing since 2005 [10]. In parallel, there are also efforts within public health services to improve cultural inclusiveness and sensitivity, including efforts to more accurately record Indigeneity within medical records [33]. However, there are no previous studies that specifically explore differences in characteristics of deliberate self-poisoning presentations associated with Indigenous cultural identity for this time period; and there has only been one study of Northern Territory hospital presentations that examined whether Indigeneity was related to clinical management of hospital-treated suicidal thoughts and behaviours, finding that Indigeneity was not a significant predictor of admission to hospital after adjustment for other relevant characteristics of patients (e.g., gender, age, residence, type of suicidal thoughts/behaviours) and their ED management and care [12]. Developing exploratory models to better understand the factors associated with decisions about aftercare for hospital-treated self-harm and whether there are differences in clinical practice associated with patient Indigeneity will assist in identifying priorities for action to ensure hospital care for self-harm can close, rather than widen or maintain, gaps in self-harm and suicide prevention outcomes.

## 2. Aims

The aims of this study were, for a cohort of deliberate self-poisoning admissions, to:

Compare the demographic and clinical characteristics of presentations by cultural identity (Indigenous or not Indigenous).

Estimate the magnitude and direction of the association between cultural identity (Indigenous or not Indigenous) and discharge destination (psychiatric hospital v other).

Explore risk factors for discharge destination (psychiatric hospital v other) within the Indigenous deliberate self-poisoning cohort.

## 3. Methods

### 3.1. Design

This study used a ten-year retrospective cohort (2003–2012) of consecutive, hospital-treated, index (first in the study period) deliberate self-poisoning admissions to the Hunter Area Toxicology Service (HATS), a sentinel unit using a case register for self-poisoning patients at the Calvary Mater Newcastle hospital. The study had ethical approval from Hunter New England Human Research Ethics Committee (17/09/4.05) and the Aboriginal Health and Medical Research Council (EC1314/17).

### 3.2. Setting

The Calvary Mater Newcastle hospital is based in Newcastle, a regional city in New South Wales, Australia. The hospital, through HATS is the primary referral centre for all deliberate self-poisoning in the Newcastle, Lake Macquarie and Port Stephens local government areas (servicing a population of approximately 385,000 people), and a tertiary referral centre for the Greater Hunter region (for an additional 170,000 people).

During the cohort period, all deliberate self-poisoning presentations to Calvary Mater Newcastle Emergency Department were admitted to HATS [34]; and all patients received a clinical assessment conducted by toxicology staff and a mental health assessment conducted by consultation-liaison psychiatry staff. The clinician who conducted the mental health assessment made determinations regarding psychiatric diagnoses and discharge destinations. Diagnoses were made based on the DSM-IV, the diagnostic manual current at the time of this study. Assessment information was recorded on a standardised, pre-formatted form and the toxicological and psychiatric data from these forms were entered prospectively into a case register dataset by trained data entry staff [35]. Cultural identity was confirmed by self-report at presentation to the hospital to administration staff as part of the initial triage process.

### 3.3. Data

The data extracted for this study included: demographic information (gender, age group, marital status, employment status, highest educational level, and postcode of residence to calculate the SEIFA index [36] of social advantage), clinical characteristics (psychiatric diagnoses reported by DSM-IV major diagnostic categories, suicidal acuity and type of psychiatric contact in previous 12 months) and service delivery information (admission and discharge time and dates, discharge destination). These variables were selected to explore differences between Indigenous and non-Indigenous participants; and to adjust for other factors associated with decisions regarding discharge destination.

Demographic characteristics were included primarily to adjust for population differences between Indigenous and non-Indigenous populations. Clinical characteristics were chosen based on factors identified as important within best-practice recommendations for clinical assessment and management [7,16] and where there is existing evidence of associations with differences in hospital care and management amongst self-harm patients [8,37].

### 3.4. Data Analysis

Descriptive statistics were used initially to explore the dataset, including checking for normality distribution, outliers, and to identify if there were any patterns that would inform further analysis or warrant data transformation [38]. A series of one-way chi-square analyses (categorical variables) were conducted to compare demographic and clinical characteristics by cultural identity (Indigenous or non-Indigenous).

To explore the wide range of predictors of discharge to the psychiatric hospital for the entire cohort, we used a series of univariate analyses to investigate the relationship between demographic and clinical variables and discharge to psychiatric hospital vs. other discharge destination (including home, police, gaol, general hospital, and death), reported as Odds Ratios with 95% Confidence Intervals (OR: 95% CI). Due to the number of comparisons, we considered statistical significance to be <0.001 (see Appendix A).

Since so little is known about Indigeneity and discharge destination, we conducted a series of exploratory analyses to determine if Indigeneity was independently associated with discharge to the psychiatric hospital. We used a univariate logistic regression to calculate the unadjusted odds for Indigeneity as a predictor of discharge to the psychiatric hospital. We then explored the independent association of Indigeneity with discharge to the psychiatric hospital in two ways, taking into the account the relatively small number of Indigenous participants discharged to the psychiatric hospital. We first selected statistically significant predictor variables from univariate analyses (Appendix A) and made a series of sequential adjustments by each variable and reported the adjusted odds ratio (AOR: CI 95%) for the Indigeneity variable. This allowed an understanding of the impact of adjusting for other predictor variables on the magnitude and direction of the relationship between Indigeneity and discharge to psychiatric hospital. Second, we developed an exploratory multivariate logistic regression model of discharge to the psychiatric hospital. We used a three-level logistic regression model with a forward elimination technique at each level to produce a parsimonious model of discharge to the psychiatric hospital; level one—demographics, level two—clinical characteristics and level three—Indigenous cultural identity, reported as Adjusted Odds Ratios with 95% Confidence Interval (AOR: CI 95%). The logistic regression model used all significant univariate predictor variables (from Appendix A) in the first two levels, after review of possible multi-collinearity, with a single variable selected where multi-collinearity existed.

A final exploratory model investigating the predictors of psychiatric hospitalisation within the Indigenous (only) cohort was also conducted. The modelling was conducted through three forced entry models limited to inclusion of three predictor variables each, because of the small number of discharges to psychiatric hospital within the Indigenous cohort, and based on the significant predictor variables within the stepwise regression. IBM SPSS Statistics 25 (Chicago, IL, USA) was used for all analyses.

An Aboriginal advisory group of local Aboriginal health and community leaders who were working together to progress Aboriginal suicide prevention in the region supported and informed the approach to this study from the point of conception. Study findings were discussed with the group, who advised on interpretation, implications and arising queries from a community perspective. This group also provided comments on the manuscript, with a standing invitation to join the research group as authors.

## 4. Results

### 4.1. Participant Characteristics

The cohort comprised 3846 participants, including 149 (4%) participants who identified as Indigenous and 3697 (96%) who did not. The Indigenous patients were younger, less likely to be married or employed, and had lower tertiary education levels. The Indigenous patients had statistically lower proportions with a Mood Disorder diagnosis; and there were no differences for the other diagnostic groups. There were also no significant differences between groups regarding level of suicidality at the time of psychiatric assessment in the general hospital or types of previous psychiatric contact in the previous 12 months. See Table 1.

### 4.2. Association between Cultural Identity and Discharge to Psychiatric Hospital

Overall, 31% (*n* = 1207) of the total sample were referred to the psychiatric hospital at discharge; 32% (*n* = 1175) for non-Indigenous and 21% (*n* = 32) for Indigenous patients. At the univariate level, Indigeneity was associated with a significantly lower likelihood of being discharged to psychiatric hospital OR 0.59 (CI 0.40–0.87). Sequential adjusted models showed little change in magnitude and direction from the unadjusted OR association between Indigeneity and discharge destination, with a range of AOR 0.54–0.66; the greatest attenuation of the association was by Mood Disorder diagnosis and the greatest strengthening of the association was by Schizophrenia and Psychotic Disorder diagnosis. See Table 2. 

The results of the multivariable model for discharge to the psychiatric hospital can be seen in Table 3. Being male, older age, homeless, higher suicidality level, a Mood Disorder diagnosis, a Schizophrenia and Psychotic Disorder diagnosis or a previous psychiatric inpatient contact in the past twelve months, were associated with a higher likelihood of being discharged to the psychiatric hospital. Identifying as Indigenous had a lower likelihood of being discharged to the psychiatric hospital AOR 0.34 (CI 0.21–0.73), after adjustment for demographic and clinical characteristics in the model.

### 4.3. Characteristics of Indigenous Patients Associated with Discharge to Psychiatric Hospital

For the Indigenous (only) cohort, older age (demographic model) and higher suicidality level and Mood Disorder diagnosis (clinical characteristics model) were significant predictors of discharge to psychiatric hospital. See Table 4. 

## 5. Discussion

This is the first study investigating the relationship between Indigeneity and discharge destination of deliberate self-poisoning admissions in Australia. Perhaps the most important and new finding of the study was that the Indigenous patients were less likely to be referred to the psychiatric hospital at the conclusion of their general hospital treatment for deliberate self-poisoning, even after adjusting for other predictors.

### 5.1. Key Findings

#### 5.1.1. Indigeneity and Referral to the Psychiatric Hospital

This finding was unexpected, although disparities in type of care offered and received by Indigenous people in Australia has been documented for other illnesses and conditions [39], including for cancer [40], cardiovascular disease [41], and kidney transplants [42,43]. We can only provide speculative explanations of what the reduced likelihood of discharge to psychiatric hospital for the Indigenous group may mean. Clinicians may have seen psychiatric hospital as less relevant or less appropriate for Indigenous patients and the alternative of community care as less restrictive, more culturally relevant and perhaps more effective or less harmful; meaning they were more likely to prioritise discharges home where possible. Similarly, patient and family preferences for outpatient options may have been greater than in non-Indigenous populations. Lack of trust and previous poor experiences with institutions (both directly and historically) may contribute to the preferences and decision-making around what is most appropriate. If not reflecting patient, family or clinician preferences or perceptions about cultural appropriateness of current inpatient settings, the alternative view is that clinicians were withholding access to the psychiatric hospital in a selective or even discriminatory way.

Thus, we cannot identify directly why Indigenous patients were discharged to the psychiatric hospital less frequently. However, the models restricted to the Indigenous participants showed that older age, higher suicidality level and a Mood Disorder diagnosis increased the likelihood of discharge to the psychiatric hospital, which is consistent with standard clinical practice for all patients. The pattern of Indigenous patients being younger and less likely to have a Mood Disorder diagnosis may account for the lower rate of discharge to the psychiatric hospital. For example, younger age may have a lower association with psychiatric hospitalisation as it may be seen as less relevant because young people are often still living with family members who can provide a safe environment or hospitalisation might be seen as potentially harmful (e.g., risk of being exposed to predatory sexual or bullying behaviour in hospital [44,45]). However, further work is needed to understand the pattern of findings.

#### 5.1.2. Why Is Discharge to the Psychiatric Hospital Important?

The differential pattern of hospitalisation may have positive or negative outcomes. The balance of benefits and harms for psychiatric hospitalisation after deliberate self-poisoning or suicide attempt has never been tested in a randomised controlled trial for suicidal outcomes, even though admission to a psychiatric unit is standard practice for patients requiring more acute support and treatment [18]. Observational studies show that psychiatric hospitalisation is associated with increased risk of suicide [46], which is likely because of a combination of confounding by indication and the harms of hospitalisation. We specifically do not know about the differential risks and benefits of psychiatric hospitalisation for Indigenous patients, and it is not an issue specifically covered in the best-practice guidelines [16]; so the finding of a lower likelihood of discharge to the psychiatric hospital for Indigenous patients may or may not reflect good clinical practice.

Thus, this finding highlights the need for better understanding about what support is needed for Indigenous people who have presented to hospital after self-poisoning or self-harm. To date, only one aftercare program has specifically been designed for, targeted towards and tested with an Indigenous group (Maori [47]). In discussing the findings of the study, representatives of the local Indigenous community raised the potential role of cultural advisors or advocates within the assessment process (as used elsewhere [48]) and emphasised the importance of understanding what discharge to the psychiatric hospital means from the perspective of the patients, their family members and clinicians at an individual level. Exploring these issues further, using qualitative techniques, will allow us to identify what changes may be required in the assessment and care planning process to ensure the best fit of aftercare is available and offered, both in the shorter and longer term. We also plan to use quantitative approaches to develop aetiological or predictive models that might help to explain some of the reasons for the lower likelihood of psychiatric hospitalisation as after-care and the relationship of discharge destination to repetition of self-poisoning and suicide mortality outcomes.

Hospital presentation is also only one component of care that is important to consider. Guidelines have been developed to guide best-practice mental health assessment of Aboriginal and Torres Strait Islander people presenting to hospital with self-harm and/or with suicidal thoughts [16]. These guidelines emphasise the importance of culturally sensitive, inclusive and competent care, and communication. The guidelines also provide advice about a comprehensive set of risks, strengths and needs that should be considered during assessment to inform decisions about self-harm and suicidal crisis aftercare. In discussing the findings of this study with Indigenous community members, they specifically highlighted the impact of inter-generational trauma, including the long-lasting effects of the Stolen Generation [49], and how these factors were frequently not recognised in delivery of routine clinical services. They identified that greater acknowledgement and consideration of the impact of inter-generational trauma may help inform what aftercare is required and suggested that timely, community-based support may avert and prevent crises before people present to hospital. Furthermore, there is a growing number of community and residential programs with promising results (e.g., [50,51,52]) that may be relevant to build from and evaluate their effectiveness within our local setting.

#### 5.1.3. Demographic and Clinical Characteristics by Cultural Identity

The other set of important findings from this study were the differences in characteristics of Indigenous and non-Indigenous patients. The Indigenous patients were younger, more likely to be single, not in the paid workforce and to have primary or secondary school as their highest educational level (with these latter characteristics potentially being driven in part by the younger age profile). The greater proportion of young Indigenous people in a hospital-treated self-harm population is consistent with official national institutional data [5] and other peer-reviewed studies that report self-harm admission patterns by Indigenous status within centres within the Northern Territory and Western Australia [12,13,15]. Gender and socio-economic status (as measured by the SEIFA) did not vary by cultural identity, with females and those in the mid-range deciles for socio-economic advantage accounting for 3 out of 5 admissions for both Indigenous and non-Indigenous cohorts. This pattern of findings for gender and socio-economic status is similarly consistent with official national institutional data [5]. Thus, these findings highlight that clinical services involved in after-care of self-poisoning patients should recognise the greater proportion of younger Indigenous patients and consider specifically how to work most effectively with Indigenous young people [16] and the nature of and access to effective youth-specific interventions including ambulatory and inpatient options [53] for Indigenous young people.

Clinical characteristics were similar across the groups, with few differences in the proportion diagnosed with a range of mental illness or substance use disorders (DSM-IV Major Groups), recognition of relationship problems (DSM-IV V codes), level of suicidality or level of previous psychiatric contact. The exception was for Mood Disorders, with the Indigenous cohort being less likely to have a Mood Disorder diagnosis. Estimates of rates of psychiatric diagnoses for Indigenous people vary widely across settings, assessment processes and types of studies. For example, in a systematic review of prevalence rates of psychiatric disorders for Australian Indigenous people, one-year estimates of the prevalence of depression ranged from 5% to 51% and varied across settings (general community, Aboriginal medical service, prison) and assessment methods [54].

There are many potential explanations for the lower prevalence of Mood Disorders for the Indigenous patients reported in this study. The finding may be accurate and valid, indicating that the context of the self-poisoning was different for the Indigenous cohort. It may also have occurred because of measurement error (clinicians failing to recognise Mood Disorder in Indigenous patients due to differing communication styles, or different features of Mood Disorder being present for Indigenous patients), response bias (patients not disclosing Mood symptoms at assessment), or a Type 1 statistical error (false rejection of the null hypothesis) because of the number of statistical comparisons. As there were no differences in diagnosis for any other mental disorders or other conditions, if a true effect, it would appear not to be a generalised diagnostic issue difficulty (presuming there are no real differences), but rather an issue specific to presentation or identification of Mood Disorders. Using a diagnostic tool validated for Indigenous populations may be an appropriate way to explore this in future studies (e.g., [55,56]) although local consultation highlighted that community members believed it was “how” questions were asked rather than “what” (the content) was asked that was the key issue (Bron Rose, 2021, personal communication).

#### 5.1.4. Generalisability of These Results and Comparability with Other Studies

Since the HATS service covers all hospital-treated cases, then the results should be generalisable to the local Hunter population. In our study, 4% of the cohort identified as Indigenous; whilst in the HATS primary catchment area, Indigenous people accounted for 4.6% of the population [57], suggesting no disproportionate presentations of Indigenous people. However, these findings are inconsistent with the higher rates of presentation by Indigenous people in other studies [10,13,15] and institutional data [5]. It is important that this finding is explored further in order to better understand whether this finding results from the local Indigenous community having lower rates of deliberate self-poisoning compared to other regions of Australia (true difference), a difference in self-poisoning *v* self-injury methods (sampling bias), a consequence of local Indigenous people being less likely to present to the hospital after a deliberate self-poisoning episode (sampling bias) or a consequence of people not reporting their Indigeneity when presenting to hospital after deliberate self-poisoning (misclassification bias). The results of this study in a largely urban regional centre are not necessarily generalisable to other populations and should be thought of as complementary rather than comparative with published studies in representative populations in the Northern Territory or remote Western Australia.

### 5.2. Implications

There are several implications arising from this study. First, the study highlights the importance of investigating patterns of clinical care and after-care for hospital-treated self-harm in Indigenous clinical populations within local services across Australia, in order to better recognise the diversity of patient needs, community characteristics, and service system responses. Second, this study highlights the importance of accurate and culturally appropriate assessments in determining aftercare offered. Third, it highlights the need to better understand what the differential pattern of referral to psychiatric hospital means for Indigenous patients, their families and the health services; as well as the longer-term outcomes associated with psychiatric hospitalisation.

During the consultation process, where interpretation and implications of the findings were discussed, representatives of the local Indigenous community asked whether it was possible to identify which staff had completed cultural competency training and whether this was associated with distinct patterns of clinical management. These community representatives felt strongly that the study highlighted the need for embedded cultural advisors within the mental health assessment and discharge process, as recommended in the Royal Australian and New Zealand College of Psychiatrists’ Position Statement [58], to help ensure treatment decisions were culturally sensitive and relevant. While upskilling and ensuring the mainstream workforce can conduct culturally sensitive assessments and authentically engage with Indigenous patients is important, growing the Indigenous clinical workforce and developing and making available culturally specific models of care were also identified as priorities to allow the health service to become culturally inclusive and to deliver culturally appropriate care more broadly.

The Indigenous community representatives also emphasised the need to ensure that local Indigenous family and community members were provided with relevant information and support, were routinely included in discharge discussions and were offered available community care given the high proportion of people discharged home from general hospital after deliberate self-poisoning. This suggestion is consistent with bi-national clinical practice guidelines [7], guidelines specific to supporting Indigenous people presenting to hospital after self-harm [16] and mental health service modelling developed through review of the literature and expert discussion [59]. Identifying the degree to which this happens routinely is important to consider.

These are all avenues for future work. We plan to conduct qualitative studies in partnership with the community to explore potential improvements in service along with quantitative studies to better understand the “reasons” for the association of Indigeneity and discharge to the psychiatric hospital. Using data linkage to track outcomes (e.g., health service use, employment, mortality) following discharge from the general hospital, with observational study designs, would also enable some understanding of the benefits and harms of psychiatric hospitalisation as an after-care intervention for non-Indigenous and Indigenous patients.

### 5.3. Limitations

This study has several limitations. It drew upon data collected during routine clinical assessments and recorded in the HATS case register. Thus, the data were clinician-derived and not based on standardised diagnostic interviews or other validated instruments. The cohort was also regionally representative only for the Hunter area, and restricted to self-harm by poisoning (not all self-harm types). Discharge to psychiatric hospital is a clear-cut outcome; however, routine clinical data within the case register does not capture the specifics or nuances of decision-making that contribute to discharge planning. Statistically, whilst the numbers for total sample size and primary outcome (psychiatric hospitalisation) were large, the sample size and primary outcome numbers for the Indigenous group were smaller and multiple comparisons were made. Indigeneity was based on self-identification and the accuracy of these data will have been affected by the degree to which patients were prepared to identify as Indigenous within this institutional context, if they had knowledge of their heritage, and whether administration staff asked about cultural identity as part of routine practice; noting that identification as Indigenous in administrative health data has been found to be historically inaccurate (underestimates) [33]. More broadly, it is also important to note that it is difficult to compare the findings of this study with other peer-reviewed studies because of differences in inclusion criteria, type of presenting self-harm and diversity of community and service system characteristics and so generalisation of these results to other populations should be done with great caution. Finally, the study time setting is representative of an earlier (important) historical period and it is unknown the degree to which the findings reflect more recent practices.

Within this context, replicating this study is a priority. Locally, the study provides a clear snapshot of previous practice, providing exploratory models of predictors of psychiatric hospitalisation as after-care and highlighting areas needing further investigation, regarding clinical decision making and effectiveness of this sort of intervention. However, despite the longitudinal design the number of Indigenous patients was relatively small meaning additional work is needed to confirm the reliability of the patterns. Geographically, the study should be replicated to identify the degree to which the pattern of care is common across hospitals or specific to this referral population and clinical model and to better understand the drivers and outcomes of psychiatric hospitalization after deliberate self-poisoning for Indigenous Australians. Comparisons with other countries with First Nations populations (e.g., New Zealand, Canada) would also be valuable.

## 6. Conclusions

The findings from this study highlight the value of conducting local investigations of hospital-based deliberate self-harm data using case registers. Indigenous patients were younger and less likely to be discharged to psychiatric hospital than the non-Indigenous patients, even after adjustments for confounding between cultural identity and other variables. Clinical services might use this information to plan for community-based after-care service provision, particularly for young Indigenous patients. This study provides a good foundation for further exploration of the issues, including the need to increase our understanding about what factors are affecting after-care decisions, through qualitative investigations; and whether the differential patterns of discharge to the psychiatric hospital represent valid and appropriate clinical management, effectiveness of the intervention for suicidal behaviour outcomes, or if changes in clinical practice are required. 

## Figures and Tables

**Table 1 ijerph-19-12238-t001:** Demographic and Clinical Characteristics by Cultural Identity.

	Non-Indigenous	Indigenous		
Variable	*n* (%) 3679 (96)	*n* (%) 149 (4)	Chi Square	*p*-Value
**Gender**				
Male	1398 (38)	59 (40)	0.19	0.660
Female	2299 (62)	90 (60)		
**Age group**				
18–25 years	943 (26)	57 (38)	16.53	0.002
26–40 years	1365 (37)	50 (34)		
41–50 years	762 (21)	30 (20)		
51+ years	627 (17)	12 (8)		
**Marital status**				
Married/de facto	1187 (32)	32 (22)	16.51	<0.001
Single/never married	1849 (51)	100 (68)		
Separated/divorced/widowed	622 (17)	16 (11)		
**Employment**				
Employed	1087 (29)	18 (12)	21.81	<0.001
Not in paid work	1419 (38)	76 (51)		
Unknown	1191 (32)	55 (37)		
**Highest education** (missing = 347)				
Tertiary	2775 (83)	9 (7)	11.05	0.001
Primary or secondary	587 (17)	128 (93)		
**SEIFA** (based on postcode) *				
Decile 1–3	812 (23)	42 (28)	19.68	0.02
Decile 4–6	2260 (62)	84 (57)		
Decile 7–10	553 (15)	19 (13)		
**DSM IV diagnoses**				
Mood disorder	1753 (47)	50 (34)	11.50	0.001
Anxiety disorder	449 (12)	24 (16)	2.09	0.149
Schizophrenia and other Psychotic disorder	228 (6)	15 (10)	3.68	0.055
Substance use disorder	1819 (49)	83 (56)	2.42	0.120
Personality disorder	686 (19)	34 (23)	1.71	0.191
Other disorders ^#^	776 (21)	31 (21)	0.00	0.957
Relationship problems (V codes) ^^^	1830 (50)	84 (56)	2.71	0.100
Multiple diagnoses	2613 (71)	111 (75)	1.01	0.315
**Suicidal level at assessment**				
No plan or ideation	2293 (62)	101 (68)	2.06	0.357
No plan; low to moderate ideation	651 (18)	23 (15)		
Active plan and/or intense ieation	753 (20)	25 (17)		
**Psychiatric contact (12 months)**				
No contact	1409 (38)	65 (44)	8.39	0.015
Outpatient contact only	1642 (44)	49 (33)		
Inpatient contact (any)	646 (18)	35 (24)		

* Higher scores/deciles = more advantage, less disadvantage relative to others. ^^^ Relational Problems: relational problem related to a mental disorder or general medical condition, parent-child relational problem, partner relational problem, sibling relational problem, relational problem NOS. ^#^ “Other” includes: Academic problem, Antisocial behaviour, Bereavement, Borderline intellectual functioning, Malingering, Medication induced movement disorder NOS, Neuroleptic induced acute akathisia, Noncompliance with treatment, Occupational problem, Phase of life problem, Physical abuse of adult/child, Sexual abuse of adult/child, Adjustment disorders, Dissociative disorders, Eating disorders, Factitious disorders, Impulse control disorders NOS, Mental disorders due to a general medical condition, Sexual and gender identity disorders, Sleep disorders, Somatoform disorders, personality disorder diagnoses.

**Table 2 ijerph-19-12238-t002:** Unadjusted and Sequentially Adjusted Estimates for Cultural Identity Predicting Discharge to Psychiatric Hospital.

	*n* (%) Discharged to Psychiatric Hospital	OR	95% CI	*p* Value
Non-Indigenous	1175 (32)		Referent category	
Indigenous	32 (21)	0.59	[0.40–0.87]	0.009
**Indigeneity association after adjusting for**		**AOR**	**95% CI**	***p* value**
Age (continuous)		0.62	[0.42–0.93]	0.020
Gender		0.58	[0.39–0.87]	0.008
Housing		0.57	[0.37–0.86]	0.008
Psychiatric contact (12 months)		0.55	[0.36–0.82]	0.004
Alcohol co-ingestion		0.59	[0.40–0.88]	0.009
Suicidal level at assessment		0.55	[0.33–0.91]	0.020
Mood Disorder		0.66	[0.44–0.99]	0.040
Schizophrenia and Psychotic Disorder		0.54	[0.36–0.81]	0.003
Relationship Problem		0.61	[0.41–0.90]	0.014

Each adjusted model is adjusted only for each single variable.

**Table 3 ijerph-19-12238-t003:** Stepwise Logistic Regression Model for Discharge to Psychiatric Hospital.

*n* = 3846 Study Cohort *n* = 1203 Discharged to Psychiatric Hospital	OR	95% CI
**Level 1 ^a^**		
*Gender*		
Female	Referent category	
Male	1.27	[1.09–1.48]
*Age*		
Continuous (year)	1.01	[1.01–1.02]
*Housing*		
Stable	Referent category	
Homeless	2.22	[1.39–3.54]
**Level 2 ^b^**		
*Suicidal level at assessment*		
No thoughts or plan	Referent category	
Low/moderate suicidal ideation	11.22	[8.82–14.27]
Suicide plan/intense suicidal ideation	46.68	[35.64–64.13]
*Mood disorder*		
No	Referent category	
Yes	2.23	[1.79–2.78]
*Schizophrenia and Other Psychotic disorder*		
No	Referent category	
Yes	4.93	[3.26–7.45]
*Psychiatric contact (12 months)*		
None	Referent category	
Outpatient only	1.15	[0.92–1.46]
Inpatient (any)	2.83	[2.10–3.82]
**Level 3 ^c^**		
*Cultural identity*		
Non-Indigenous	Referent category	
Indigenous	0.34	[0.21–0.73]

^a^. Variable(s) entered in Block 1: Age, Gender, Employment, Marital status, Housing, Education, Postcode. ^b^. Variable(s) entered in Block 2: Suicidal level at assessment, Mood Disorder, Psychotic Disorder, Anxiety Disorder, Substance Use Disorder, Relational Problems, Any Other Disorder, Multiple Diagnoses, Psychiatric Contact, Alcohol Co-ingestion. ^c^. Variable(s) entered in Block 3: Cultural Identity.

**Table 4 ijerph-19-12238-t004:** Exploratory Logistic Regression Models for Discharge to Psychiatric Hospital for Indigenous Cohort.

Variables	OR	95% CI [Lower–Upper]
Indigenous cohort: *n* = 145; *n* = 32 discharged to psychiatric hospital ^>^		
**Model 1: Demographics** ^>a^		
*Age*	1.05	[1.01–1.08]
**Model 2: Clinical characteristics** ^>b^		
*Suicidal level at assessment*		
No thoughts or plan	Referent category	
Low/moderate suicidal ideation	26.26	[5.99–115.25]
Suicide plan/intense suicidal ideation	79.86	[17.10–372.99]
*Mood disorder*		
No	Referent category	
Yes	5.62	[1.68–18.80]
**Model 3: Other characteristics** ^>c^	Nil significant predictors	

^>^ Indigenous multivariate analysis predictor variables entered: ^>a^—Age, Gender, Housing; ^>b^—Suicidal level at assessment, Mood Disorder, Psychosis Disorder; ^>c^—Alcohol co-ingestion, Relational Problems, Past Type of Psychiatric Contact.

## Data Availability

The data presented in this study are available on request from the corresponding author. The data are not publicly available due to ethical restrictions on who can access and use the data.

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
