# Peer review of "Indigeneity and Likelihood of Discharge to Psychiatric Hospital in an Australian Deliberate Self-Poisoning Hospital-Treated Cohort"

_ijerph, 2022, doi:10.3390/ijerph191912238_

Round 1
Reviewer 1 Report
This is a well written and excellent study into self-harm by Indigenous peoples within a particular region.
I have a few minimal suggestions:
- eliminate the 'has/ have been' constructions. This construction in some instances could be replaced with are, is, or was. For instance on page 2 "The impact of colonization on Australian Indigenous communities is significant" is far more accurate than the 'has been'. Page 3, bottom paragraph contains multiple has/have/had been constructions.
- page 5 - sentence starting with 'Because'. The authors' could use 'Since' instead of 'Because'
- page 7 "predatory sexual ..." this sentence has an awkward reading. Particularly the 'predatory sexual' being left without a completing word whilst be followed by drug misuse. I would suggest 'predatory sexual and bullying behaviours as well as drug misuse.' Albeit, I recognize that this might not exactly capture the authors' intent.
- Lastly, I would suggest that the authors expand the study beyond the borders of Australia to include a specific region with the U.S. and Canada.
-
Reviewer 2 Report
This manuscript exploited a retrospective clinical cohort of Indigenous Australians to study the likehood of psychiatric hospital admission and deliberate self-harm. The introduction and methods are well described, but the presentation of the results need to be improved. Here are some major/minor points:
1. the Indigenous cohort is relatively small compared with non-Indigenous cohort, thus the understanding of different factors contribute to the admission of hospital or self-harm/suicide will be limited.
2. the font size/type is not consistant in the manuscript, eg."Discussion", "key findings"... also the tables have different formatting as well.
3. Please follow the formating requirements of the journal for the reference
